# Steroid hormones sulfatase inactivation extends lifespan and ameliorates age-related diseases

Mercedes M. Pérez-Jiménez [1], José M. Monje-Moreno [1], Ana María Brokate-Llanos[1],
Mónica Venegas-Calerón [2], Alicia Sánchez-García[2], Paula Sansigre[1], Amador Valladares[1],
Sara Esteban-García[3], Irene Suárez-Pereira[3,7], Javier Vitorica[4,5,6], José Julián Ríos [2], Marta Artal-Sanz [1],
Ángel M. Carrión[3] & Manuel J. Muñoz [1]✉

Aging and fertility are two interconnected processes. From invertebrates to mammals, absence of the germline increases longevity. Here we show that loss of function of *sul-2*, the *Caenorhabditis elegans* steroid sulfatase (STS), raises the pool of sulfated steroid hormones, increases longevity and ameliorates protein aggregation diseases. This increased longevity requires factors involved in germline-mediated longevity (*daf-16*, *daf-12*, *kri-1*, *tcer-1* and *daf-36* genes) although *sul-2* mutations do not affect fertility. Interestingly, *sul-2* is only expressed in sensory neurons, suggesting a regulation of sulfated hormones state by environmental cues. Treatment with the specific STS inhibitor STX64, as well as with testosterone-derived sulfated hormones reproduces the longevity phenotype of *sul-2* mutants. Remarkably, those treatments ameliorate protein aggregation diseases in *C. elegans*, and STX64 also Alzheimer's disease in a mammalian model. These results open the possibility of reallocating steroid sulfatase inhibitors or derivates for the treatment of aging and aging related diseases.

[1] Centro Andaluz de Biología del Desarrollo (CABD)—Universidad Pablo de Olavide (UPO), Departamento de Biología Molecular e Ingeniería Bioquímica, UPO/ CSIC/JA, Sevilla, Spain. [2] Instituto de la Grasa (CSIC), Campus Universitario Pablo de Olavide (UPO), Sevilla, Spain. [3] Departamento de Fisiología, Anatomía y Biología Celular, Universidad Pablo de Olavide (UPO), Sevilla, Spain. [4] Departamento de Bioquímica y Biología Molecular, Facultad de Farmacia, Universidad de Sevilla, Sevilla, Spain. [5] Instituto de Biomedicina de Sevilla (IBiS)-Hospital Universitario Virgen del Rocío/ CSIC/Universidad de Sevilla, Sevilla, Spain. [6] Centro de Investigación Biomédica en Red sobre Enfermedades Neurodegenerativas (CIBERNED), Madrid, Spain. [7] Present address: Europsychopharmacology and psychobiology research group, Universidad de Cadiz, CIBERSAM, INiBICA, Cadiz, Spain. ✉email: mmunrui@upo.es

Animals can extend lifespan by activating different genetic pathways. This increase of longevity is a regulated process that relay in the coordination of different tissues and environmental signals. Hormones are key players in tissues and cell communication. Consequently, they are involved in different pathways that regulate longevity, among those insulin and insulin-like growth factor, transforming growth factor-β (TGF-β) or dafachronic acids (DAs) which are described to affect lifespan at least in the model organism *Caenorhabditis elegans*[1]. Gonad is an endocrine tissue that produces steroid hormones to regulate different physiological aspects of the organism, including longevity. In *C. elegans*, germline ablation extends lifespan by non-completely understood mechanisms. Several factors are needed for the increase in longevity, including synthesis of DA by the somatic gonad as well as the transcription factor encoded by *daf-16*, homologue to the human FOXO, and the nuclear receptors encoded genes *daf-12*, *nhr-80* and *nhr-49* (refs. [2,3]).

The classical function of steroid hormones is considered to be the activation of hormones receptors to transcribe their target genes. Steroid hormones are not only produced in gonads but also in other tissues. Those produced in the nervous system are known as neurosteroids[4]. Neurosteroids, in addition to bind to hormone receptors, modulate neurotransmission either through direct interaction with neurotransmitter receptors or by other mechanisms[5].

Steroid hormones can be sulfated by a sulfotransferase enzyme, generating a profound change in the chemical features of the hormone that impairs its function as hormone receptor activator. Those sulfated hormones are considered to be an inactive reservoir of hormones that can be activated upon removal of the sulfate moiety by the activity of hormone sulfatases[6]. Sulfated steroid hormones can also be active as neurosteroids, regulating neurotransmission[5].

Some sulfated steroid hormones, like dehydroepiandrosterone sulfate (DHEAS), have long been related to aging. The level of this hormone declines with age and in age-related diseases such as sarcopenia or Alzheimer's disease (AD), which has generated the speculation of a causative effect[5,6].

Here we show that inhibition of the steroid sulfatase generates an increase of the percentage of sulfated hormones and, associated with that, an increase in longevity and the improvement of the symptoms related to protein aggregation diseases. This increase in longevity is mainly dependent on the same factors described for longevity caused by germline ablation. Treatment with STX64, a specific inhibitor of the steroid sulfatase enzyme, mimics the beneficial effects in longevity and protein aggregation diseases observed in the mutant. Interestingly, treatment with STX64 also ameliorates the cognitive symptoms and plaque formation in a mammalian model of AD. Finally, the observed phenotypes are recapitulated by treatment with sulfated C19 androgen steroid hormones but not with the non-sulfated forms or the sulfated C21 pregnenolone hormone, indicating that the causative beneficial effect of *sul-2* inhibition is due to the increase of sulfated C19 steroid hormones rather than reduction of the non-sulfated form. This work suggests that STX64 or sulfated C19 steroid hormones could be a possible treatment for aging and/or aging-related diseases.

## Results

### Identification of *sul-2* as a regulator of longevity.
Unravelling new elements that govern the genetic control of aging is key to improve our understanding of this intricate biological process and improve human healthspan. To this aim, we isolated *Caenorhabditis elegans* thermotolerant mutants using a previously reported protocol[7] and identified an allele *pv17* of the *sul-2* gene, which encodes one of the three members of the *C. elegans*

sulfatase family[8]. Worms carrying either the isolated (*pv17*) allele or the null allele (*gk187*) of *sul-2* live longer than wild type, although the *gk187* allele shows a bimodal curve with a sub-population that has an early mortality (Fig. 1a, b and Supplementary Fig. 1a–c. Additional data of all longevity assays in Supplementary Dataset 1). Pumping frequency and mobility during aging declines similar or slower than wild type, overall for the *pv17* allele, suggesting a healthier lifespan (Supplementary Fig. 1d, e). Deletion of the other two sulfatases genes, *sul-1* and *sul-3*, does not increase lifespan, indicating that the sulfatase *sul-2* has a main role in the regulation of longevity (Supplementary Fig. 1f). Mutations in *sul-2* also enhance the developmental phenotypes of mutants in the insulin/insulin-like growth factor (IGF) receptor *daf-2* (Supplementary Fig. 1g–i), which was used for gene mapping and identification. The *pv17* allele introduces a single amino acid substitution (G46D) resulting in a reduction of function phenotype. The curated sequence slightly differs from the one published[9] (Supplementary Fig. 2).

### *sul-2* encodes a sulfatase of steroid hormones.
Sulfatases are a large protein family involved in different biological processes and with a wide range of substrates. The placement of curated *sul-2* in the sulfatases phylogenetic tree is uncertain, but when compared to mammalian sulfatases, *sul-2* clusters closer to the Arylsulfatases type H, F, E, D and the steroid sulfatase type C (Fig. 1c) which probably originated from a common ancestor gene[8]. *sul-1* and *sul-3* belong to a different family of sulfatases (Fig. 1c). We hypothesized that *sul-2* may exert its activity by modifying sulfated steroid hormones. Steroid hormone sulfatases are conserved proteins that participate, among other processes, stimulating proliferation in hormone-depending cancers[6]. Specific inhibitors for this type of enzymes have been generated, such as STX64[10], which has been used to treat patients with hormone-depending cancers[11]. We treated wild-type animals with STX64 and observed an increase in longevity (Fig. 1d and Supplementary Fig. 3a–c). STX64 treatment also phenocopies other *sul-2* mutant phenotypes (below and Supplementary Fig. 3d). STX64 does not further increase the longevity of *sul-2* deletion mutants, suggesting STX64 increases longevity is by inhibiting the sulfatase activity of SUL-2 (Fig. 1d).

We measured sulfated steroid hormones levels by a high-resolution HPLC-TOF-MS in *sul-2* mutant and found a higher proportion of sulfated hormones in this strain as compared to wild-type worms (Fig. 1e and Supplementary Table 1). All these data suggest that SUL-2 can act as a steroid hormones sulfatase and regulate longevity through the alteration of the sulfated state of one or several steroid hormones.

### *sul-2* and gonadal longevity share genetic factors.
To investigate if *sul-2* acts in a known longevity pathway, we performed genetic interaction studies with known alleles that affect longevity. Mutations in the IGF receptor *daf-2* increase lifespan through the transcription factor DAF-16/FOXO[12]. *sul-2* mutations further extend the lifespan of *daf-2* reduction of function mutants (Fig. 2a and Supplementary Fig. 4a, b), suggesting that *sul-2* acts in a different pathway to regulate longevity. However, the increased longevity of *sul-2* mutants is mainly suppressed by DAF-16 loss of-function (Fig. 2b and Supplementary Fig. 4c, d).

Longevity conferred by lack of germline also requires DAF-1[12,13], which translocates to the nuclei mainly in intestinal cells. However, in insulin signalling mutants, DAF-16 localizes to the nucleus of most cells[14]. In *sul-2* mutants, DAF-16 localizes predominantly to intestinal nuclei (Supplementary Fig. 5a, b), suggesting a role for *sul-2* in germline-mediated longevity. Other essential factors for germline-mediated longevity, such as the intestinal ankyrin-repeat protein *KRI-1/KRIT-1* and the transcription elongation factor

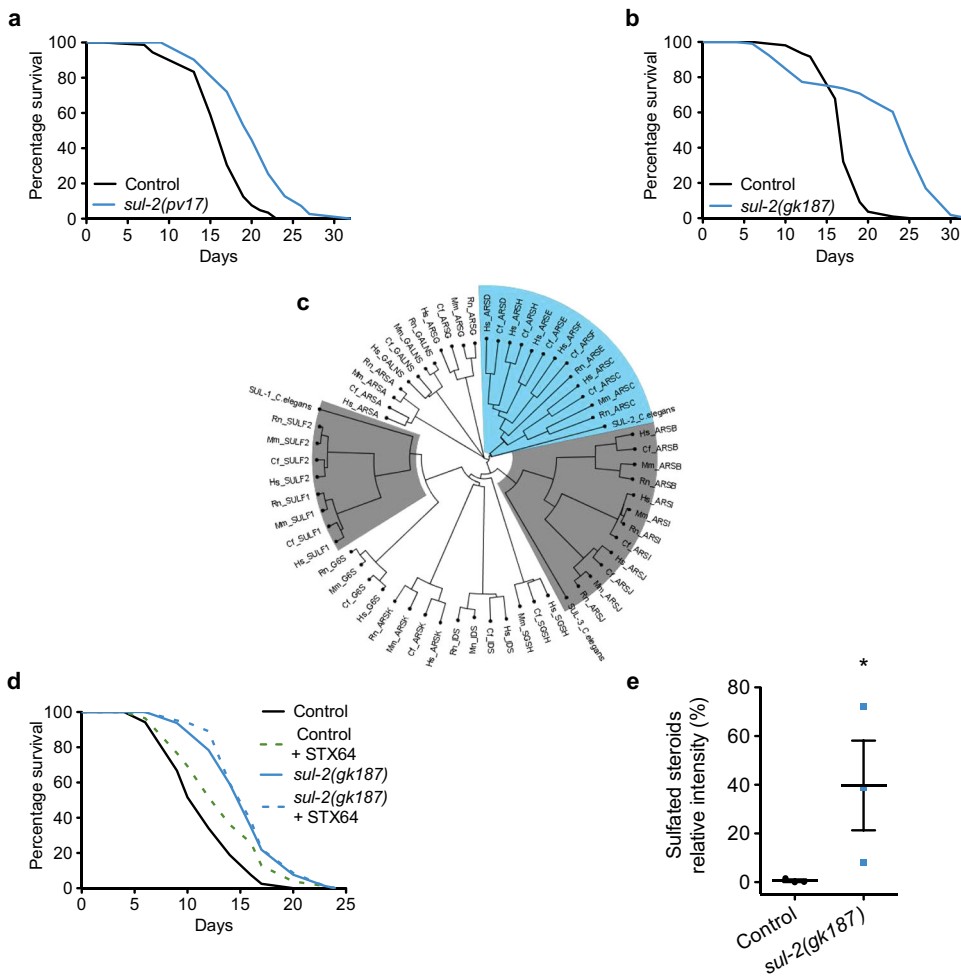

**Fig. 1 sul-2 activity reduction increases longevity and the levels of sulfated steroid hormones. a** *sul-2(pv17)* point mutation allele lives longer than wild type. **b** The null allele *sul-2(gk187)* also increases lifespan. **c** Phylogenetic tree of mammalian sulfatases and the three *C. elegans* sulfatases. Note that SUL-2 clusters with steroid sulfatases (type C) among others, in blue. Phylogenetic relation with other *C. elegans* sulfatases are indicated in grey. **d** Inhibition of steroid hormones sulfatase by STX64 (1 μg/ml) increases lifespan in wild-type animals, but not in *sul-2(gk187)* background. Worms were cultivated in UV-killed *E. coli*. **e** Deletion of *sul-2* generates an increase in the percentage of steroid hormones in the sulfated stage. Data from three independent assays are displayed; $n \approx 100,000$ worms per sample. Mean ± SEM; one-tailed Mann–Whitney $t$-test; $^*p = 0.050$. Relative quantification of hormones from independent experiments are shown in Supplementary Table 1. Statistics and additional longevity data are shown in Supplementary Dataset 1. Source data are provided as a Source Data file.

*TCER-1/TCERG1* are also required for a fully increased longevity of *sul-2* mutants, although a mild effect is observed with the nuclear hormone receptor NHR-80 and no effect is observed in NHR-49, which are also required for germline-mediated longevity[15–17] (Fig. 2c–f and Supplementary Fig. 4e–g). Moreover, deletion of *sul-2* has little effect on the longevity of the germline-less mutant *glp-1* or *mes-1*[18] (Fig. 2g, h). However, we do observe an additive effect in the reduction of function allele *pv17* in a *glp-1* background (Supplementary Fig. 4h), which may suggest some additional effect of the mutated protein. All these data suggest that *sul-2* mediates signalling from the gonad to regulate longevity. Interestingly, *sul-2* mutations do not affect fertility, reproductive age or gonad morphology (Fig. 2i, j and Supplementary Fig. 5c–e). Taken together, our findings suggest that *sul-2* affects a signal that regulates longevity to adjust lifespan to the reproductive status without affecting gonadal function. Similar to germline ablation[19], we also observed that *sul-2* mutants increase further the longevity in dietary restriction, suggesting that *sul-2* is not implied in this intervention that affects longevity (Fig. 2k).

In germline-less animals, the activation of the nuclear receptor DAF-12 by bile acid-like steroids called DAs triggers an increase in longevity[20]. We observed that *daf-12* is also needed for the increased longevity of *sul-2* mutants (Fig. 2l and Supplementary Fig. 4i), indicating that *sul-2* inactivation causes the alteration of the sulfated steroid hormones pool, generating a signal upstream of DAF-12 that imitates the longevity signals of gonad depleted animals. DAF-36 converts cholesterol to 7-dehydrocholesterol in the first step of the biosynthetic pathway of $\Delta^7$-DA[21,22]. Therefore, DAF-36 is also needed for the increased longevity of germline-less animals[23]. Similarly, DAF-36 is required for the longevity conferred by the steroid sulfatase inhibitor STX64 (Fig. 2m), placing the signal generated by sulfated steroid hormones upstream of or parallel to the biosynthesis of DAs.

**sul-2 is expressed in sensory neurons.** We have studied the anatomical location of *sul-2* expression from an extra-chromosomal array and in single-copy insertion transgenic strains. We found that *sul-2* is expressed only in a few sensory neurons, mainly in the amphids ADF and ASE, and phasmids PHA and PHB. There is no detectable expression in the germline in any transgenic strains (Fig. 2n and Supplementary Figs. 6 and

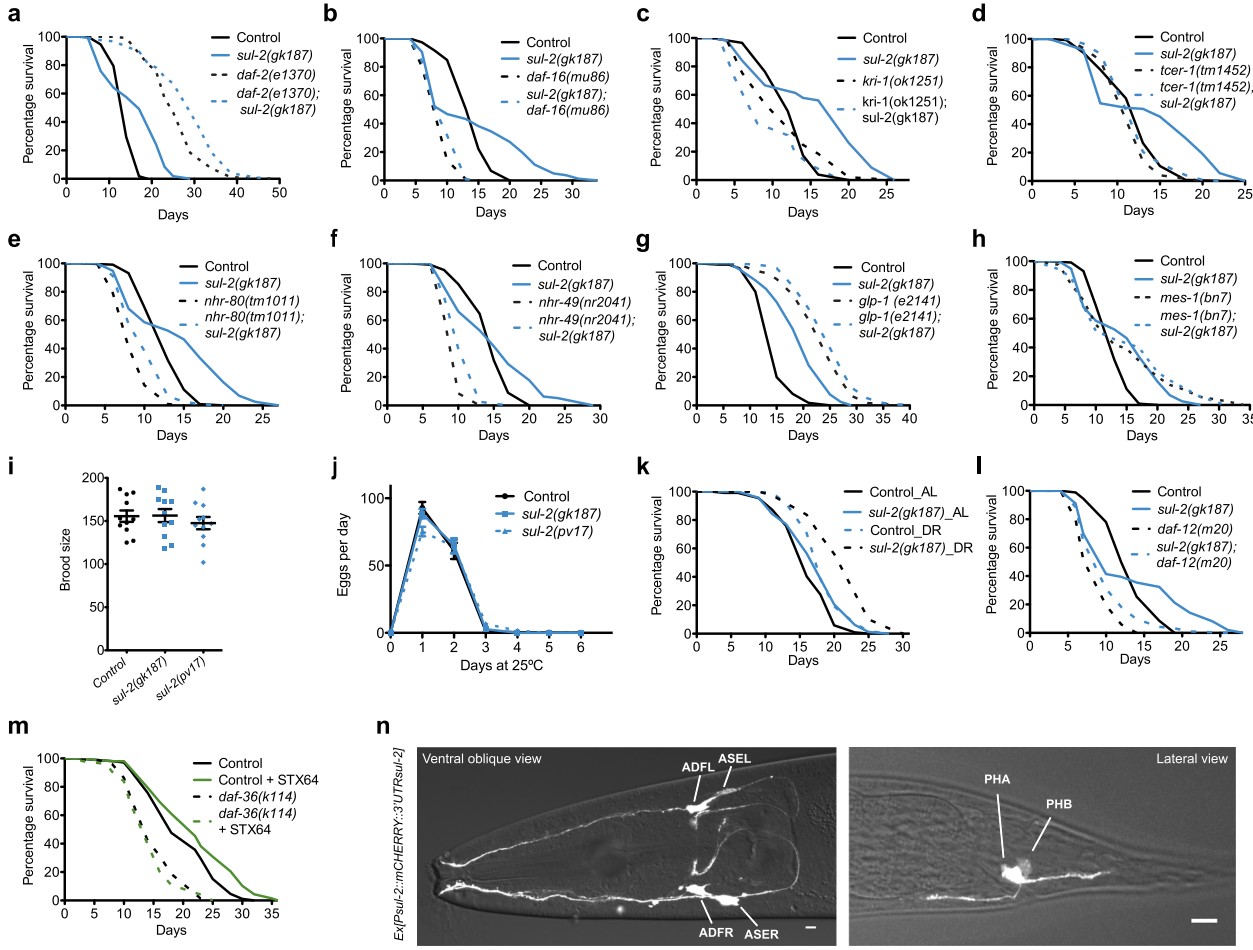

**Fig. 2 Genetic interactions and cellular location of *sul-2* expression.** Genetic analysis show that *sul-2* mimics most of the genetic interaction described for animals without germline, but do not affect fertility. **a** *sul-2* enhances longevity in *daf-2(e1370)* background. **b** *daf-16* transcription factor is required for *sul-2* longevity. **c–e** The essential factors for germline-loss longevity *kri-1(ok1251)*, *tcer-1(tm1452)*, and partially *nhr-80(tm1011)* are required for *sul-2* longevity. **f** *nhr-49(nr2041)* does not suppress *sul-2* increased lifespan. **g, h** *sul-2* deletion has not significant increase of longevity in germline-less mutants *glp-1(e2141)* and *mes-1(bn7)*. **i, j** Brood size of *sul-2* mutants are not significantly different to wild type (one-way ANOVA test; n.s.) and reproductive period is not affected. Data from an independent experiment (25 °C) of two are shown; *n* = 11 *per* strain. Mean ± SEM. **k** Dietary restriction conditions (DR) enhances longevity in *sul-2* deletion background. **l** *daf-12* transcription factor is required for *sul-2* longevity. **m** The Rieske-like oxygenase *daf-36* is necessary for the increase of longevity upon inhibition of steroid hormone sulfatase, STX64 (1 μg/ml). **n** *sul-2* is transcriptional expressed in sensory neurons, mainly ADF and ASE in the head, and PHA and PHB in the tail. Scale bar represents 10 μm. Statistics and additional longevity data are shown in Supplementary Dataset 1. Source data are provided as a Source Data file.

7a, b). ASE neurons are responsible for the attractive response of Na⁺ and Cl⁻, among others[24]. Defects in odour sensing affect longevity[25]. Therefore, we tested the ability of *sul-2* mutants to respond to Cl⁻ or Na⁺ and found no differences compared to wild-type animals (Supplementary Fig. 7c, d). Furthermore, *sul-2* mutation increases the longevity of *daf-10(m79)*, a long-lived mutant defective in sensory cilia formation[25] (Supplementary Fig. 7e). These results show that the longevity phenotype observed in *sul-2* mutants is not due to the impaired functionality of sensory neurons.

**Reduction of *sul-2* improves aging-associated diseases.** Aging is considered the main risk factor for the onset of neurodegenerative disorders like Parkinson, Huntington or ADs. These disorders are caused by the progressive decline of proteostasis, which results in protein aggregation that compromises cellular functions and finally causes cell death[26]. Germline-deficient *C. elegans* delay the symptoms derived from the proteotoxicity of ectopically

expressed human β-amyloid (βA)[27]. We tested if *sul-2* mutations or STX64 treatment improves the symptoms of *C. elegans* models for neurodegenerative diseases. In a *C. elegans* Parkinson's disease model, in which human α-synuclein expression in muscle cells causes age-dependent paralysis[28], *sul-2* mutation or treatment with STX64 significantly improves mobility (Fig. 3a, b, Supplementary Fig. 8a–c and Supplementary Videos 1–4). Loss of function of SUL-2 decreased the number of α-synuclein aggregates (Supplementary Fig. 8d, e), suggesting better handling of protein aggregates in worms with reduced steroid sulfatase activity. To further assay the neuroprotective effect of reduced *sul-2* activity, we tested a strain expressing α-synuclein in dopaminergic neurons. In this model, GFP-labelled dopaminergic neurons die due to α-synuclein toxicity[29]. Consistently, *sul-2* mutants show increased neuron survival compared with control worms, indicating a neuroprotective action of reduced steroid-hormone sulfatase activity (Fig. 3c, d). In a Huntington neurodegenerative model expressing polyglutamine repeats fused to YFP, a construct that aggregates in adult worms[30], we found that

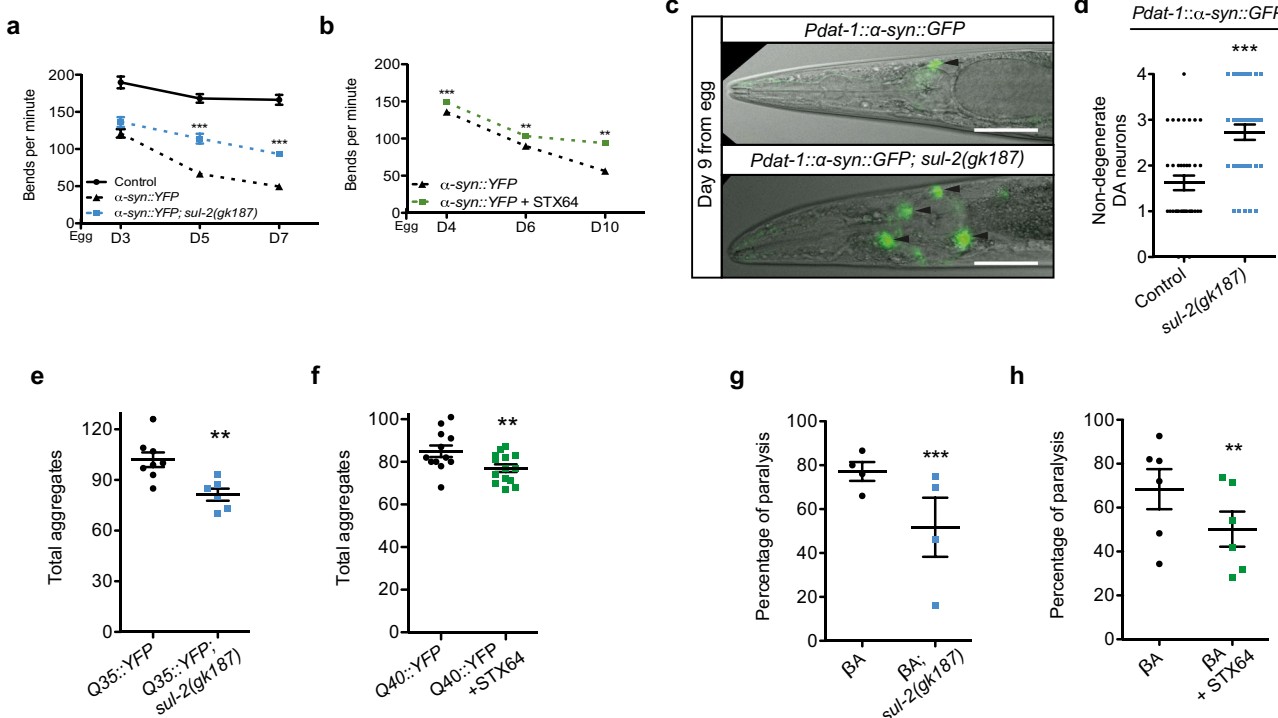

**Fig. 3 Reduction of steroid hormone sulfatase activity ameliorates proteotoxicity in *C. elegans*. a** NL5901 strain expressing α-synuclein in muscle cells reduce mobility with age at a slower rate in a *sul-2(gk187)* background; $n \approx 17$ per sample **b** or under treatment with STX64 (1 μg/ml); $n \approx 9$ per sample. One of three independent biological replicates is represented in **a** and **b**, two-tailed Mann–Whitney *t*-test. Additional replicates assays are shown in Supplementary Fig. 8a, c. **c, d** Neurodegeneration of dopaminergic (DA) neurons in UA44 strain expressing human α-synuclein is reduced in *sul-2(gk187)* background. Animals at day 9 were imaged and the number of normal DA neurons were counted; a representative image of each condition and quantifications from two biological replicates are displayed; $n = 37$ considering the total number of worms analysed per strain. One-tailed Mann–Whitney *t*-test. Scale bar represents 50 μm. **e** Q35 aggregates are reduced in *sul-2(gk187)* background at 8-day-old worms; $n \approx 6$ per strain **f** or by treatment with STX64 (1 μg/ml) in Q40 background at 5-day-old worms; $n \approx 12$ per strain. One-tailed unpaired *t*-test in **e** and **f**. **g** Expression of human β-amyloid in muscle provokes a thermodependent paralysis in L4-young adult stage in GMC101 strain; the paralysis phenotype is ameliorated in *sul-2* deletion mutant. Data display the percentages from four independent biological replicates; $n = 125$ per strain **h** or by STX64 (1 μg/ml) treatment. Data display the percentages from six independent biological replicates; $n \approx 215$ per strain. $\chi^2$ test was used to analyse the data in **g** and **h**. In all graphs mean ± SEM are displayed. **$p \leq 0.01$, ***$p \leq 0.001$. Exact *n* and *p* value are included in Source Data file.

both *sul-2* mutation and treatment with STX64 reduce the number of aggregates (Fig. 3e, f). We also tested two different strains of AD in worms, where immobility is caused by expression of βA protein in muscle cells[31,32]. Consistently, *sul-2* mutation and STX64 treatment delay paralysis (Fig. 3g, h and Supplementary Fig. 8f, g). All these data show that inhibition of *sul-2* protects the nematode against aging-related proteotoxicity.

**sul-2 inhibition improves Alzheimer in a mammal model.** As STX64 ameliorated neurodegeneration in *C. elegans* models, we tested the effect of this drug on cognitive alterations provoked by intrahippocampal βA oligomers infusion, an acute AD mammalian model (Fig. 4a). Previously, it has been reported that local administration of DU-14, an inhibitor of steroid hormones sulfatase, could alleviate memory loss caused by intrahippocampal administration of βA oligomers in a mammalian model[33]. We observed that both local and systemic STX64 treatments revert the cognitive deficiencies, measured by passive avoidance test, caused by intrahippocampal administration of βA oligomers (Fig. 4b). To evaluate the effect of STX64 oral treatment on amyloid pathology in a chronic AD mice model, we assessed the effect of 3–4 weeks of STX64 oral treatment on amyloid deposition in the neocortex (the cerebral cortex and the hippocampus) of >15-month-old APP-PS1

mice (Fig. 4c). At this age corresponding to a late stage of amyloid deposition in the neocortex of the APP-PS1 model, the analysis of βA plaque density and size revealed a significant reduction, except for plaque size in the hippocampus, in mice treated with STX64. Moreover, βA immunoreactive area is reduced in both tissues (Fig. 4c–f). Interesting, when we compared βA deposition in old (>15-month-old) APP-PS1 mice treated with STX64 with the normal temporal course of amyloid deposition in un-treated APP-PS1 mice, we observed that STX64 reduces βA deposition in APP-PS1 mice older than 15 months compared to that observed in APP-PS1 mice of 10–12 months of age (Fig. 4g). All these results indicate that STX64 treatment in APP-PS1 mice reduces βA deposition. We wondered if this histological improvement correlated with amelioration of cognitive behavioural deficit. For that, we compared cognition capacities in APP-PS1 mice older than 15-month-old treated with vehicle or STX64 during 3–4 weeks. While vehicle-treated APP-PS1 mice showed a deficit in passive avoidance test (Fig. 4h), those mice treated with STX64 completely reverted cognitive deficiencies, reaching similar levels to <15-month-old wild-type mice. All these results point out that the alterations in βA metabolism provoked by STX64 reduce the cognitive behaviour deficiencies induced by βA accumulation in acute and chronic AD mice models, suggesting a potential for STX64 as a pharmacological therapy against neurodegenerative diseases.

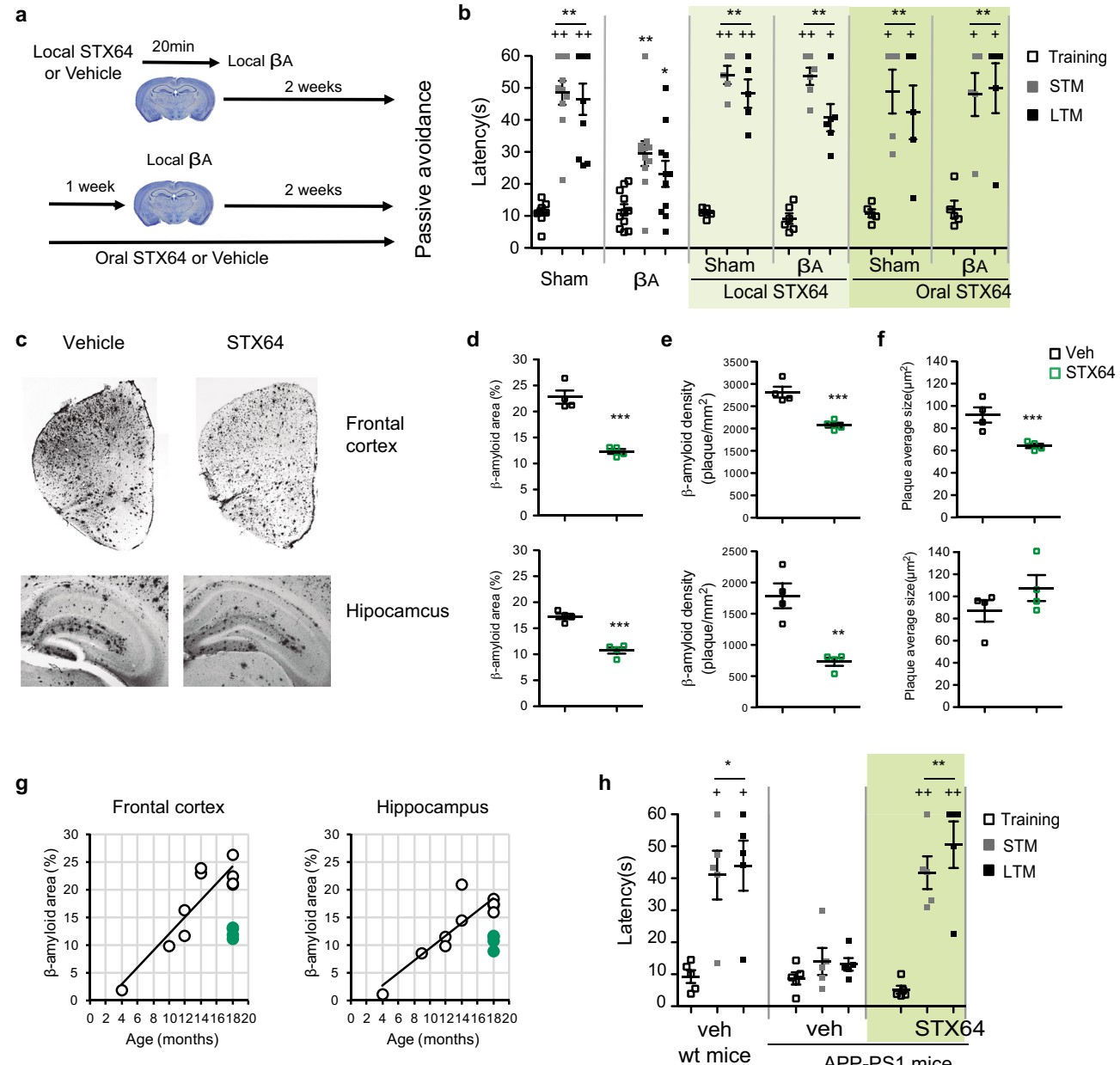

**Fig. 4 Reduction of steroid hormone sulfatase activity ameliorates proteotoxicity in murine models. a**, **b** The effect of intrahippocampal and oral administration of STX64 in the passive avoidance test in wild-type mice injected with β-amyloid oligomers in the hippocampus; $n \approx 5$ mice per groups. Two-tailed Tukey's test. **c** Representative β-amyloid-immunoreactive images assessed in the frontal cortex and the hippocampus of APP-PS1 mice older than 15 months of age after 3–4 weeks of vehicle or STX64 intake (0.005 mg/ml in drink water). **d–f** Quantification of the percentage of β-amyloid area (**d**), deposition density (**e**) and average plaque size (**f**) in the frontal cortex and the hippocampus of >15-month-old APP-PS1 mice after 3–4 weeks of oral administration with STX64 or vehicle. $n = 4$ mice per groups. Two-tailed unpaired $t$-test. **g** Temporal course of β-amyloid deposition in APP-PS1 mice and the effect of 3–4 weeks STX64 oral treatment on β-amyloid area in the frontal cortex (upper panel) and the hippocampus (lower panel). The number of microphotographs used was more than three in all the mice used. **h** Effect of oral administration with STX64 in more than 15-month-old APP-PS1 mice, compared with APP-PS1 and wild-type mice older than 15 months in the passive avoidance test. $n = 5$ mice per groups. Two-tailed Tukey's test. In histological analysis, * represents significant differences between vehicle and STX64 administrated APP-PS1 mice. In behavioural test, * represents significant differences between the short-term and long-term memory sessions (STM and LTM, respectively) and the training session in the same experimental group; + represents significant differences between the STM and LTM sessions between each experimental group and β-amyloid group. A symbol, $p < 0.05$, two symbols, $p < 0.01$, and three symbols, $p < 0.001$. In all graphs mean ± SEM are displayed. Exact $n$ and $p$ value are included in Source Data file.

**Sulfated C19 androgen hormones recap *sul-2* mutant phenotypes.** In mammals, sulfated hormones have been long considered inactive forms that function mainly as reservoirs and are activated by steroid sulfatases[6], although a direct action of sulfated hormones in the reproductive and the nervous system has been observed[34,35]. In this last tissue, those hormones are named neurosteroids and their main function is the modulation of neurotransmission[5]. In order to sort out whether the beneficial effect of *sul-2* inhibition is due to the reduction of non-sulfated hormones or the increase of sulfated hormones, we tested the

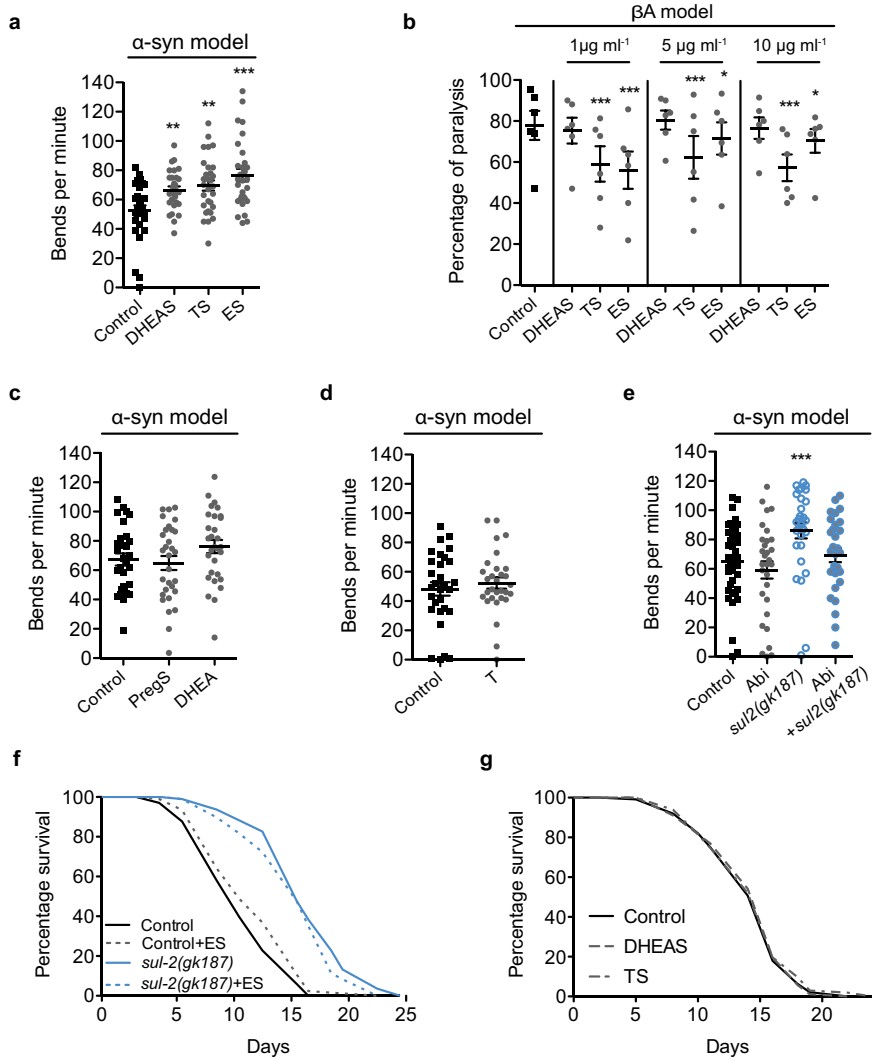

**Fig. 5 Sulfated C19 androgen steroid hormones mimic *sul-2* inactivation. a** NL5901 strain expressing α-synuclein in muscle cells reduce mobility with age. Treatment with DHEAS, TS or ES improves the mobility (1 μg/ml). Data from three biological replicates are displayed, n ≈ 30 worms per sample. Two-tailed Mann–Whitney *t*-test. **b** Paralysis of GMC101 Alzheimer's disease model is ameliorated with TS and ES. Data display the percentages from six independent biological replicates; n ≈ 200 per sample. χ² test. Additional assays in regular NGM plates are shown in Supplementary Fig. 9a. **c, d** No effect is observed with non-sulfated steroid hormone DHEA or testosterone (T), neither with sulfated C21 steroid hormone, pregnenolone sulfate (PregS) in Parkinson's model, NL5901 strain (1 μg/ml). Data from three biological replicates are displayed, n = 30 worms per sample. **e** Treatment with abiraterone (Abi) (1 μg/ml) does not affect mobility phenotype of NL5901 strain, but suppresses the beneficial effect of *sul-2* deletion allele. Data from at least three biological replicates are displayed, n ≈ 30 per sample. Two-tailed Mann–Whitney *t*-test in **c** and **d**. **f** Treatment with ES (1 μg/ml) increases lifespan in a wild-type background but does not further increase in *sul-2(gk187)*. **g** DHEAS or TS (1 μg/ml) does not affect longevity. Additional biological replicate assays are shown in Supplementary Fig. 9c and d. Statistics of longevity assays are shown in Supplementary Dataset 1. In all graphs mean ± SEM are displayed and \*p ≤ 0.05, \*\*p ≤ 0.01, \*\*\*p ≤ 0.001. Exact *n* and *p* value are included in Source Data file.

commercially available sulfated steroid hormones that are highly presented in the mutant (Supplementary Table 1). We observed that the C19 androgens DHEAS, testosterone sulfate (TS) and epitestosterone sulfate (ES) improve the mobility in the Parkinson model of *C. elegans*, with a better result for ES (Fig. 5a). Similar results are obtained in the Alzheimer model with TS and ES but not with DHEAS (Fig. 5b and Supplementary Fig. 9a). Non-sulfated DHEA or non-sulfated testosterone show no effect, neither pregnenolone sulfate, which belongs to the C21 group of steroid hormones (Fig. 5c, d). These results indicate that at least some sulfated C19 androgens are involved in the protective effect against protein aggregation diseases and strongly suggest that the beneficial effect of *sul-2* inhibition is due to the increased levels of this type of hormones. In agreement with these results, treatment with the antiandrogenic compound abiraterone (Abi)[36], does not

affect wild-type animals but suppresses the beneficial effect of *sul-2* mutation (Fig. 5e).

We then tested if those hormones are also involved in the other phenotypes observed in the *sul-2* mutant. Treatment with any of those sulfated hormones generate an increase of L1 arrest in a *daf-2(e1370)* background as observed in *sul-2* or STX 64 treated animals (Supplementary Fig. 9b). Interestingly, treatment with ES, but not TS or DHEAS, increases longevity on a wild-type background and did not further increase longevity in *sul-2* mutants backgrounds (Fig. 5f, g and Supplementary Fig. 9c, d), indicating that both interventions share the same molecular mechanism. Thus, addition of ES recapitulated all the phenotypes described for *sul-2* inhibition and strongly suggesting that the causative effect of *sul-2* mutation is due to the increase of C19 androgen sulfated hormones related to ES.

## Discussion

Gonad is a key tissue in the regulation of lifespan. Germline regulates longevity by inhibiting the production of DA in the somatic gonads. Consistently, germline ablation or mutations that abolish the generation of germline increase lifespan by activation of DA synthesis[2]. Gonads are also the classical tissue that produces sex steroids, although is not the only one. Our data indicate that inhibition of the sulfatase activity either by mutation or by STX64 raises the level of sulfated steroid hormones, which in fact generate an increase in longevity. This increase in longevity depends on common factors involved in lifespan extension produced by germline loss, suggesting that both processes are in fact linked. We cannot distinguish whether the prolongevity effect of sulfated steroid hormones participates in the same pathway or acts in parallel to the germline longevity sharing some element of this pathway. The fact that *sul-2* inhibition does not depend on NHR-49, or only partially depends on NHR-80, which are essentials for germline-mediated longevity[2,3], point to the second option. We have also studied the level of sulfated steroid hormones in the germline-less *glp-1* mutant, and we do not observe an increase in sulfated hormones (Supplementary Table 2). Those data favour the idea that gonads produce steroid hormones, which are modified by sulfation. These sulfated steroid hormones, probably altering neurotransmission, produce an increase in longevity, through common factors to germline-less animals. The fact that the enzymes involved in the sulfate modification of steroid hormones (sulfatase SUL-2 and sulfotranferase SSU-1) are expressed in sensory neurons[37,38] suggests that alteration of the sulfated state of hormones may acts in the integration of environmental cues, such as nutrient availability, with the reproductive status, which are two-linked processes[39,40] (see the model in Supplementary Fig. 10).

In *C. elegans*, cell proliferation of the somatic cells only occurs during development and in larval stages but not in the adult stage; therefore, increasing longevity is due to the maintenance of the postmitotic cells. One of the stresses observed in *C. elegans* adult cells is the aggregation of endogenous proteins, which generates cellular misfunction[41]. This age-related formation of aggregates is also observed upon ectopic expression of aggregation-prone proteins, like β-amyloid or α-synuclein[42]. Long-lived mutants such as *daf-2* or *glp-1* delay the aggregation toxicity through a different mechanism including chaperon expression and degradation by proteasome or autophagy[43]. We have shown that inhibition of the sulfatase activity or treatment with sulfated C19 androgen hormones impinge not only in longevity but also reduce protein aggregation and its toxic consequences in *C. elegans* models of protein aggregation diseases (see model in Supplementary Fig. 10).

Regulation of steroid hormones by sulfation is a conserved process. In mammals, sulfotransferases and sulfatases are expressed in different tissues, including the nervous system, similar to what we observe in *C. elegans*[44]. In humans, C19 steroid hormones have also been involved in longevity. For instance, DHEAS declines with age and has been used as a marker of aging, raising speculations of a causative effect on sarcopenia, poor cognitive function and other aging-associated diseases[6] including AD[45]. Our data show that inhibition of the steroid sulfatase by mutation or by STX64 treatment extends lifespan in *C. elegans* and protects against aging-associated proteotoxicity in nematodes. Interestingly, similar effects are observed upon treatment with sulfated C19 steroid hormones. With the method used, due to probable modifications of the hormone once is incorporated, it is difficult to assert which are the specific hormone/s that are in fact causing this effect. We speculate that it should be a sulfated androgen C19 steroid, closely related to ES, which has shown the most robust effect. In this sense, it is

intriguing that in humans, women are more prone to suffer AD[46,47]. Both, sulfated C19 steroid hormones or inhibitors of steroid hormone sulfatase, are of potential interest for the treatment of aging and aging-associated diseases. Sulfated C19 steroid hormones do not easily pass the blood–brain barrier[5]. We demonstrate that oral treatment of STX64 alleviates the symptoms of AD in a murine model, indicating that STX64 passes the blood–brain barrier. STX64 has been tested in phase II clinical assay for the treatment of hormone-dependent cancer and did not show serious toxic effects[11]. This suggests that inhibitors of steroid hormone sulfatase, such as STX64 could be pharmacological compounds of interest to be reallocated for treating aging and aging-associated diseases.

## Methods

**Strains.** All the strains were grown under standard conditions[48]. N2: wild-type Bristol isolate, GM11: *daf-16(m26)I;fer-15(b26)II*, GMC101: dvIs100[*Punc-54:: amyloid-β1-42:3′UTR unc-54 + Pmtl-2::GFP]II*, *GM88: *daf-16(m26)I;sul-2(pv17) V*, *GM369: *kri-1(ok1251)I*, *GM354: *kri-1(ok1251)I;sul-2(gk187)V*, *GM363: *daf-16(mu86)I; muIs109[Pdaf-16::GFP::DAF-16 cDNA + Podr-1::RFP]X*, *GM365: *daf-16(mu86)I;sul-2(pv17)V;muIs109*, *GM366: *daf-16(mu86)I;sul-2(gk187)V;muIs109*, *GM425: *nhr-49(nr2041)I (2)*, *GM426: *nhr-49(nr2041)I (1)*, *GM427: *nhr-49 (nr2041)I; sul-2(gk187)V (2)*, *GM430: *nhr-49(nr2041)I; sul-2(gk187)V(1)*, *GM436: *nhr-49(nr2041)I; sul-2(pv17)V (1)*, *GM437: *nhr-49(nr2041)I; sul-2(pv17) V (2)*, AM140: *rmIs132[Punc-54::Q35::YFP]I*, *GM387: *rmIs132;sul-2(gk187)V*, AM141: *rmIs13[Punc-54::Q40::YFP]*, EG4322: *ttTi5605II;unc-119(ed9)III*, *GM410: pvIs8[Psul2::mCHERRY::3′UTR sul-2]II*, CL2006: dvIs2[pCL12(Punc-54::Aβ1-42::3′ unc-54) + pRF4(rol-6(su1006)]II*, *GM392: dvIs2;sul-2(gk187)V*, DH26: *fer-15(b26) II*, GM6: *fer-15(b26)II; daf-2(e1370)III*, GM270: *fer-15(b26)II; daf-2(m577)III*, *GM125: *fer-15(b26)II;sul-2(pv17)V*, *GM325: *fer-15(b26)II;sul-2(gk187)V*, *GM293: *fer-15(b26)II;daf-2(m577)III;sul-2(pv17)V*, *GM361: *fer-15 (b26)II; sul-1 (gk151)X*, *GM431: *fer-15(b26)II; sul-3(tm6179)X*, CF2167: *tcer-1(tm1452)II*, *GM348: *tcer-1(tm1452)II;sul-2(gk187)V*, *GM353: *tcer-1(tm1452)II;sul-2(pv17)V*, GM63: *daf-2(e1370)III*, DR1942: *daf-2(e979)III*, *GM134: *daf-2(e1370)III;sul-2 (pv17)V*, *GM291: *daf-2(e1370)III;sul-2(gk187)V*, *GM215: *daf-2(m577)III;sul-2 (pv17)V*, *GM229: *daf-2(e2979)III;sul-2(pv17)V*, *GM407: *glp-1(e2141)III*, *GM408: glp-1(e2141)III;sul-2(gk187)V*, NL5901: pkIs2386[unc-54p::α-synuclein::YFP + unc-119(+)]III*, *GM379: pkIs2386;sul-2(gk187)V*, BX165: *nhr-80 (tm1011)III*, *GM360: nhr-80(tm1011)III;sul-2(gk187)V*, *GM358: *nhr-80(tm1011)III;sul-2(pv17)V*, PR678: *tax-4(p678)III*, NY2067: *ynIs67[Pflp-6::GFP]III;him-5(e1490)V*, *GM314: ynIs67[Pflp-6::GFP]III;pvEx3*, QZ65: *daf-10(m79)IV*, *GM351: *daf-10(m79)IV;sul-2 (gk187)V*, *GM126: *sul-2(pv17)V*, *GM371: *sul-2(gk187)V*, *GM420: *sul-2(gk187) V*; dvIs100[Punc-54::amyloid-β1-42:3′UTR unc-54 + Pmtl-2::GFP]II*, *GM272: *sul-2(pv17)V;daf-12(m20)X*, *GM429: *sul-2(gk187)V; daf12(m20)X* *GM364: *sul-2 (gk187)V; mes-1(bn7)X*, DR20: *daf-12(m20)X*, *GM359: *mes-1(bn7)X*, *GM430: sul-3(tm6179)X*, *GM370: *sul-1(gk151)X*, VZ155: *oyIs26[Pops-1::GFP]X*, *GM318: oyIs26[Pops-1::GFP]X;pvEx3*, *GM288: pvEx1[Psul-2::mCHERRY::3′UTR sul-2 + pGK10(sca-1::GFP)]* *GM303: N2;pvEx3[Psul-2::mCHERRY::3′UTR sul-2]*, GR1366: mgIs42 [tph-1::GFP + pRF4(rol-6(su1006)]*, *GM320: mgIs42;pvEx3*, UA44: baln11 [Pdat-1::α-Syn::unc-54 3′-UTR; Pdat-1::GFP]*, *GM391: *fer-15(b26)II; sul-2(gk187)V; baln11*, * Indicate the strains backcrossed with our N2 or generated during the study.

**Mutant isolation.** In order to obtain the *sul-2(pv17)* mutant strain we performed a mutagenesis protocol. Briefly: The temperature-sensitive fertilization-defective mutant *fer-15(b26ts)*, was treated with 25 mM ethyl methanesulfonate (EMS)[49]. Sets of 30 mutagenized L4 larvae were incubated for 8 days at 15 °C in tubes containing 6 ml S medium with *Escherichia coli*[50]. F2 eggs were purified by alkaline hypochlorite treatment to obtain synchronous L1 larvae[51]. Approximately 10,000 synchronous F2 or F3 L1 larvae were harvested from each tube and submitted to thermal stress at 30 °C for 7 days on agar plates spread with OP50. To ensure independence of the mutants, only one survivor per tube was saved after confirming L1 thermotolerance at 30 °C. One of those mutants isolated was *sul-2 (pv17)* allele.

**pv17 identification.** To map the mutation *sul-2(pv17)*, we used an snp-SNP-based method with the Hawaiian-type strain, combined with classical genetic markers[52]. Using this approach, we found that *sul-2(pv17)* is located between SNP CE5-168 at 1.00 cM and SNP pkP5062 at 1.11 cM on chromosome V, equivalent to a physical distance of 205 kb. Then, using transgenic lines we tried to complement the mutation with genes of this region, and found that the wild-type *sul-2* gene complemented the mutant *sul-2(pv17)*. Sequencing of the wild type and *sul-2(pv17)* allele identified a mutation in the gene, which consists in a missense mutation that change a glycine to an aspartic acid residue at the position 46. An EST clone of the *sul-2* gene (yk387h10) was sequenced. The protein obtained differs from the one published in wormbase ([https://wormbase.org/species/c_elegans/gene/](https://wormbase.org/species/c_elegans/gene/)

WBGene00006309#0-9f-10], Section: Legacy manual gene description) (see Supplementary Fig. 2 and Supplementary Table 3 for primer details).

**Drug treatments**. STX64, DHEAS, dehydroepiandrosterone (DHEA), TS, Testosterone (T), ES, pregnenolone sulfate (PregS) and abiraterone (Abi) have been used in the different assays. In order to confirm that the effect of STX64 in the nematode is due to the drug and not because of the bacterial metabolism effect[53] we used non-UV and UV-coli. For the rest of the compounds NGM without peptone and with $10^{10}$ E. coli cell per plate were used[54], unless otherwise indicated. All the compounds were dissolved in DMSO and added on top of the plates after coli lawn. Final concentrations of compounds in plates are indicated in the figures.

**Statistics and reproducibility**. GraphPad Prism 5 (Version 5.0a) was used to analyse the data from C. elegans and Microsoft Excel 2016 from mice. Specific statistical test used are indicated and included in Supplementary Dataset 1 and Source Data file. Experiments yielding quantitative data for statistical analysis were performed independently at least twice, all with similar results. Micrographs images shown in the figures are representative of three independent experiments, all with similar results.

**Lifespan assays**. Strains were synchronized by hypochlorite treatment of gravid adults, grown up during two generations at 16 °C. F2 L4 animals were shifted to 25 °C, unless otherwise indicated ($t = 0$). During the first week animals were transferred to a fresh plate every 2 days, further they were transferred at least once per week and scored every 2 days until death. Animals lost or dead by non-physiological causes were censured. Survival curves were generated using the product-limit method of Kaplan and Meier. The log-rank (Mantel–Cox) test was used to evaluate differences in survival and p-values lower than 0.05 were considered significant. For STX64, DHEAS, TS and ES lifespan assays starting at L4 stage, control and treated plates were prepared fresh and animals transferred every 2 days for STX64 and every 3 days for sulphated hormones. Controls of all lifespan assays were fer-15(b26) mutant background, unless otherwise indicated, to avoid progeny production. fer-15(b26) mutation does not have effect on lifespan[55]. For lifespan details see Supplementary Dataset 1.

**Pumping assays**. Worms were growth at 16 °C until L4, and then shifted to 25 °C considering this Day 0 of adulthood. The pumping was counted under the Leica stereoscope at ×100 magnification for 30 s.

**Dauer and L1 arrest assays**. Animals were grown up to L4 at 16 °C, then shifted at 25 °C, and progeny were scored or imaged after 72 h. Photographs were taken in a Leica scope.

**Phylogenetic analysis**. Phylogenetic analysis was based on the C. elegans and mammal sequences described in Sardiello et al. (2005)[8] and multiple sequence alignments were performed by MAFFT[56], which is available at [http://www.ebi.ac.uk/Tools/mafft/]. Default parameters were used. Phylogenetic tree was visualized using the programme FigTree v.1.4.3 [http://tree.bio.ed.ac.uk/software/figtree/].

**Steroid purification and analysis**. In all, ~100,000 worms were collected in a polypropylene Falcon tube. The worms were homogenized in M9 buffer/methanol (2/3, v/v). For steroid quantification 11-deoxycortisol-2,2,4,6,6-d5 (710784; for unconjugated steroids), sodium pregnenolone-17A, 21,21,21-d4 sulfate (721301; for sulfated steroids) and diethylhexyl phthalate-d4 (DEHP-d4) were introduced into the extract. The sample was then sonicated in an ultrasonic bath for 5 min, left overnight at room temperature and centrifugated at 3000g for 5 min. The organic phase was collected and the rest of the extract residue washed again with 5 ml of methanol containing 1% acetic acid and centrifuged. The two organic phases were pooled and evaporated to dryness at 40 °C under a gentle stream of nitrogen, taken up in 1 ml of acetonitrile. The sample was purified by vortex-mixing for 1 min and centrifuged at 10,000g for 10 min. The supernatant was transferred to another 1.5 ml Eppendorf tube and evaporated to dryness under nitrogen atmosphere at 40 °C. The residue was reconstituted in 150 µl of acetonitrile or methanol–water (5:95, v/v). A 10 µl aliquot of the solution was injected into the LC–MS/MS system for analysis.

The liquid chromatograph in the HPLC/ESI-TOF-MS system was Dionex Ultimate3000RS U-HPLC (Thermo Fisher Scientific, Waltham, MA, USA). Chromatographic separation was performed as follows. The eluent components were 0.1% (v/v) formic acid in water (A) and 0.1% (v/v) formic acid in methanol (B). The proportion of B was increased from 50 to 75% in 12 min and held 25 min, then increase to 95% in 5 min and held for 5 min. Initial conditions were reached in 5 min and the equilibrium time was 2 min. The injection volume was 30 µl and the flow rate was 1 ml/min. A stainless steel column (20 × 0.46 cm i.d.), packed with 3 µm C18 Spherisorb ODS-2 (Teknokroma, Barcelona, Spain) was used. A split post-column of 0.4 ml/min was introduced directly on the mass spectrometer electrospray ion source. Mass spectrometry was performed using a micrOTOF-QII High-Resolution Time-of-Flight mass spectrometer (UHR-TOF) with q-TOF geometry (Bruker Daltonics, Bremen, Germany) equipped with an electrospray ionization (ESI) interface. All data were used to perform multitarget-screening using TargetAnalysis™ 1.2 software (Bruker Daltonics, Bremen, Germany). Collision energy was estimated dynamically based on appropriate values for the mass and stepped across a ±10% magnitude range to ensure good quality fragmentation spectra. The instrument control was performed using Compass 1.3 for micrOTOF-QII + Focus Option Version 3.0.

The in-house mass database created ex professo comprises monoisotopic masses, elemental composition and, optionally, retention time and characteristic fragment ions if known, for steroids and their derivatives compounds (Supplementary Dataset 2). For calculation of % of sulfated hormones, DA were not taken into consideration. Data evaluation was performed with Bruker Daltonics DataAnalysis 4.1. From the HPLC/TOF-MS acquisition data, an automated peak detection on the EICs expected for the $[M + H]^+$ ions of each compound in the database was performed with Bruker Daltonics TargetAnalysis™ 1.2 software. The software performed the identification automatically according to mass accuracy and in combination with the isotopic pattern in the SigmaFit™ algorithm. This algorithm provides a numerical comparison of theoretical and measured isotopic patterns and can be utilized as an identification tool in addition to accurate mass determination. The calculation of SigmaFit values includes generation of the theoretical isotope pattern for the assumed protonated molecule and calculation of a match factor based on the deviations of the signal intensities. Only those hits with mass accuracy and SigmaFit values within the tolerance limits, which were set at 5 ppm and 50, respectively, are included in the final report list that was carried out using a Microsoft EXCEL-based script. The interpretation of the MS spectra was performed using the SmartFormula3D™ module included in the DataAnalysis software. Based on expected chemistry, carbon elements, hydrogen, oxygen, nitrogen, bromine and iodine were permitted. Sodium and potassium were also included for the calculation of adduct masses. The number of nitrogen atoms was limited to an upper threshold of ten. The number of rings plus double bonds was checked to be chemically meaningful (between 0 and 50). For each steroid compound detected in the sample, the module shows the original MS and MS-MS data as peak lists. From all possible formulae for the precursor ion, only one should fit with the elemental composition expected for the $[M + H]^+$ ion and satisfy thresholds for mass accuracy and SigmaFit values. Once the correct formula is selected, the module displays the formulae and neutral losses in the MS-MS spectrum fitting to the boundary conditions for the precursor ion, and they should be consistent with the MS-MS data peak list. The SmartFormula3D checks the consistency highlighting the monoisotopic peaks with formula suggestion and the related isotopic peaks. Based on this combined data evaluation, fragmentation pattern for each steroid hormone can be generated to support its identification in the sample.

**Brood size**. Animals were grown at 16 °C. For the assays, ≈10 individual L4 animals of each strain were incubated at 20 or 25 °C, and monitored during all reproductive period. Animals were transferred to a fresh plate every single day and numbers of eggs were counted until animals did not lay more eggs.

***sul-2* expression**. To create the DNA construction for the transgenic strains, we amplified 1.7 kb upstream of sul-2 start codon (see Supplementary Table 3, primers Psul-2) named Psul-2 fragment and 0.3 kb downstream of sul-2 stop codon (see Supplementary Table 3, primers 3UTR_sul-2 + BamH) named 3′UTR sul-2, from wild-type DNA. Both regulatory elements were transitory cloned into pGEM-T Easy Vector (Promega), next inserted into the pCR™-Blunt II-TOPO™ Vector (Invitrogen) containing mCHERRY (plasmid generated and kindly provided from P. Askjaer's group, used previously to generate pBN1 (ref. [57])). Psul-2 introduced upstream of mCHERRY into ApaI/PstI and 3′UTR sul-2 downstream in BamHI site (pPV5). Extrachromosomal strains were generated by microparticle bombardment[58] of N2 with pPV5 together with pGK10 as cotransformation marker (GM288: pvEx1) and by microinjection of plasmid pPV5 into N2 (GM303: pvEx3). We observed the same expression pattern in all transgenic strains. For identification of sul-2 neurons adults animals staining with FiTC or crossed with specific neuronal marker strains were imaged by Confocal Microscope Leica SP2-AOBS. To generated the integrated strain GM410 we amplified the whole casette Psul-2::mCHERRY::3′UTR sul2 from pPV5 and inserted it, first in pGEM-T Easy Vector, next into SbfI/SphI of pBN8 recombination plasmid modified for Mos1-mediated single-copy insertion (MosSCI) into chromosome II[57] (pPV7). The transgenic animals were generated by microinjection of pPV7 with pBN40, pBN41 and pBN42 as green co-markers and pJL43.1 for transposase[59,60]. Integrated animals were selected as wild-type locomotion and non-co-markers expression individuals. Pictures of adult animals were taken by a Fluorescence Microscopy Zeiss Axio Imager M2.

**Thrashing assay**. For thrashing assays, nematodes were synchronized at 20 °C. Thrashing were assayed at the indicated days. Drug treatments were applied at egg stage until 7-day-old worm at 20 °C. For each thrashing assay, a nematode was placed in a drop of M9 buffer, let 30 s for accommodation and then the number of bends were counted during 1 min. We considered a bend when the head of the nematode crosses the longitudinal axis. Number of animals assayed per day and

condition are indicated in figure legends. Two-tailed Mann–Whitney $t$-test was used to analyse the data; $*p \leq 0.05$, $**p \leq 0.01$, $***p \leq 0.001$.

**Dopaminergic neurodegeneration induced by α-synuclein.** Synchronized nematodes were grown up to day 9 (6 days of adulthood) and imaged by the Fluorescence Microscopy Zeiss Axio Imager M2. For quantification of non-degenerate dopaminergic (DA) neurons we considered as normal neuron those where cell body and neurites are present, criteria based on Harrington et al. (2011)[61]. One-tailed Mann–Whitney $t$-test was used to analyse the data; $***p \leq 0.001$.

**Quantification of aggregates.** Animals were cultivated at 20 °C and imaged by a Leica Fluo III stereoscope or Fluorescence Microscopy Zeiss Axio Imager M2. For polyQ strains and conditions, manual quantification of total aggregates of the whole animal have been done. To quantify α-synuclein aggregates total number of fluorescent aggregates between the two pharyngeal bulbs were counted. One-tailed unpaired $t$-test was used to analyse the data; $*p \leq 0.05$, $**p \leq 0.01$.

**Paralysis assay.** For GMC101, worms were raised until L4-young adult stage at non-restrictive temperature, then shifted to 25 °C. Paralysis were calculated after 16–18 h at 25 °C. $\chi^2$ was used to analyse the data, $*p \leq 0.05$, $**p \leq 0.01$, $***p \leq 0.001$. For CL2006, paralysis was monitored during adulthood ($t0 =$ first day of adult). For each test condition, ≈50 nematodes were analysed. Nematodes were synchronized and assayed at 20 °C. For both strains, paralysis was considered when nematodes did not move forward after stimulation with a platinum pick.

**Mice strains and conditions.** The male Swiss (CD1) and APP-PS1[62] mice used in this study were purchased from an authorized provider (University of Seville, Spain) and they were habituated to standard animals housing conditions for 2–3 weeks (a 12 h light/dark cycle, temperature and humidity). Behavioural studies were performed with 8 weeks aged Swiss mice and >15-month-old APP-PS1 mice in C57Black background. For histological studies, male APP-PS1 mice from 2 to >15-month-old were used. All experiments were performed in accordance with the European Union guidelines (2010/63/EU) and with Spanish regulations for the use of laboratory animals in chronic experiments (RD 53/2013 on the care of experimental animals: BOE 08/02/2013), and the approval of the University Pablo de Olavide animal care committees was obtained prior to performing this study.

**Mice local drug infusion.** Mice were anaesthetized with 4% chloral hydrate (10 μl/kg of body weight, i.p.) and when fully anaesthetized, they were situated in a stereotactic frame. In order to injure the hippocampus, 0.5 μl of 5 μM solution of β-amyloid (βA) oligomers were injected bilaterally into the dorsal hippocampus of the mice at the following stereotactic coordinates: AP = −2.2 mm, ML = ±1.5 mm, V = −1.5 mm from the Bregma. The mice were then allowed to recover for at least 15 days. Those mice that received also STX64, 20 min before βA oligomers were administrated with 0.5 μl of STX64 (1 mg/ml) were infused in the same rostral hippocampus coordinates. STX64 and βA oligomers were delivered at a rate of 0.2 μl/min through an injection syringe (Hamilton), and left in place for 2.5 min following infusion.

**Mice oral STX64 administration.** STX64 was dissolved in drinking water at 0.005 mg/ml. Mice were exposed to STX64 solution during 3–4 weeks and the water intake was registered every day during the treatment. The estimated STX64 doses were between 1 and 2 mg/kg of mice and day.

**Step-through passive avoidance test.** Mice have an innate preference for dark and enclosed environments. During the habituation phase, mice were handled and allowed to move freely for 1 min in a chamber (47 × 18 × 26 cm, manufactured by Ugo Basile) that is divided symmetrically into one light and one dark compartment (each measuring 28.5 cm × 18 cm × 26 cm). During the training phase, mice were briefly confined to the light compartment and then 30 s later, the door separating the dark–light compartments was opened. Once mice entered the dark compartment, the door was closed automatically and the mice received an electrical stimulation (0.5 mA, 5 s and 0.3 mA, 5 s for Swiss and C57Black mice, respectively) delivered through the metal floor. In the retention tests performed at the times indicated, mice that recalled the electrical shock experience when re-placed in the light compartment avoided, or at least took longer, to enter the dark compartment. Thus, the latency to enter into the dark compartment (escape latency) is a measure of information learning or memory retention depending on how long after the training session the test was carried out. Escape latency (s) in the training, short- and long-term memory (STM and LTM, respectively) sessions are represented.

**Immunohistochemistry and histological analysis.** For immunohistochemistry, an antibody against βA (1:3000, Clone BAM-10, Sigma Aldrich Ref: A3981) was used. Antibody staining was visualized with $H_2O_2$ and diaminobenzidine, and enhanced with niquel. To minimize variability, at least five sections per mice were analysed under a bright-field DMRB RFY HC microscope (Leica). In each section, the percentage area occupied by βA, the density and the average size of βA

accumulations were quantified using Image-J software (downloaded as a free software package from the public domain: [http://rsb.info.nih.gov/ij/download.html]).

**Reporting summary.** Further information on research design is available in the Nature Research Reporting Summary linked to this article.

## Data availability

New sequence data are available in GeneBank with the access number GeneBank MW145131. Source data is provided with this paper. The source data underlying Figs. 1e, 2i–j, 3a, b, d, e, f, g, h, 4b, d–f, h, 5a, b, c, d, e and Supplementary Figs. 1d, e, g–h, 5b, c–d, 7c–d, 8a, b, c, e, f–g, 9a, b are provided as a Source data file. Other data supporting the findings of this study are available within the paper and the Supplementary Information files, or available from the authors upon reasonable request.

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

## Acknowledgements

We thank the CGC. A. Miranda-Vizuete and J. Alcedo for providing strains; Y. Kohara and P. Askjaer for DNA clones; A. Garzón, P. Askjaer and M. Vanghell for critical review of the manuscript; A. P. Pulido for bioinformatic advices; V. Carranco for his excellent technical assistance during whole project, and also A. López, A. Cano, V. Rubio, S. Romero and E. Gara for their technical support; and K. Garcia for microscopy assistance. This work was supported by the Junta de Andalucía Project P07-CVI-02697, "Fondo Europeo de Desarrollo Regional" (FEDER) and the "Consejería de Economía, Conocimiento, Empresas y Universidad, Junta de Andalucía" Project UPO-1266266 and the European Research Council ERC-2011-StG-281691. We also acknowledge the support from the Centro Andaluz de Biología del Desarrollo (CABD).

## Author contributions

M.M.P.-J. and M.J.M. conceived and designed the study and the *C. elegans* experiments. M.M.P.-J. performed most of the *C. elegans* experiments. J.M.M.-M. isolated *pv17*. M.M.P.-J., J.M.M.-M. and A.M.B.-L. constructed double mutants and did longevity curves. M.M.P.-J., P.S. and A.V. did the neurodegeneration assays in *C. elegans*. A.M.C., S.E.-G. and I.S.-P. performed mice assays. A.M.C. conceived the mice experiments. J.V. provided transgenic mice and oligomeric βA. M.M.P.-J., M.V.-C., A.S.-G. and J.J.R. performed hormone identification assays. M.M.P.-J., A.M.C. and M.J.M. performed formal analyses and data presentation. M.M.P.-J., M.A.S., A.M.C. and M.J.M. interpreted results and wrote the manuscript.

## Competing interests

Four patents of the use of steroid sulfatase inhibitors and sulfated steroid have been issued with M.J.M. and M.M.P.J., and M.J.M., M.M.P.J. and A.C. as inventors. The other authors declare no competing interest.
