## [Peer Review File · Nature Communications]

Reviewers' comments:

Reviewer #1 (Remarks to the Author):

In the manuscript, Perez-Jimenez and colleagues launched an investigation of the role of *C. elegans* sul-2 in longevity. The authors reported that loss of function of sul-2 led to increased lifespan in worms. Since sul-2 encodes a sulfatase, the authors hypothesized that sul-2 promoted longevity by modifying sulfated steroid hormones. In support of their hypothesis, the authors showed that pharmacological inhibition of sulfatase activity with a human cancer drug STX64 increased sulfated hormones and reproduced the longevity phenotype in worms. These authors continued to use sophisticated genetic tools to demonstrate that sul-2-mediated increase in longevity resembled that in mutant worms lacking germline signaling. Surprisingly, sul-2 deficiency didn't affect fertility, reproductive age or gonadal function, suggesting that one or more of the sul-2 targets can modulate longevity without impairing reproductive functions. Finally, the authors found that sul-2 deficiency improved symptoms of neurodegenerative models in worms and in mice.

The research subject is of great interest. Moreover, the findings that targeting sul-2 pathway can be beneficial in worm and murine models of neurodegenerative diseases are quite novel and potentially important. That said, the current manuscript has its limitations. First, without the knowledge of the specific steroid hormones underlying the increased longevity, the study lacks mechanistic insight. It is particularly hard to interpret how sul-2 expression in few sensory neurons could impact longevity at the organismal level. Does sul-2 homolog in mammals have a similar expression pattern in the nervous system? The authors cited one paper of STS deficiency in humans. Is there any evidence that such a deficiency in humans or in rodents increases a healthy lifespan? Second, lifespan is used as a functional readout for most of the genetic experiments without necessary information regarding altered physiologies (e.g. metabolism, intake, mobility) leading towards longevity. Finally, sul-2's effects on longevity and neurodegenerative diseases could involve independent steroid hormones and pathways. Therefore, the findings that inhibiting sul-2 ameliorated symptoms of neurodegenerative models could be the subject of an independent study. Although the data presented in the paper are quite exciting, the therapeutic potential of the drugs like STX64 is dampened by the concern that multiple steroid hormones are affected producing both beneficial and untoward effects.

Reviewer #2 (Remarks to the Author):

By quantifying lifespan in *C. elegans* in which the steroid sulfatase sul-2 gene is disabled, the authors provide evidence that sul-2 limits lifespan. Interestingly, sul-2 is expressed only a few specific

sensory neurons suggesting the desulfation of one or more steroids in those neurons is responsible for limiting lifespan. The authors also find that sul-2 mutant worms are protected against proteotoxicities of alpha-synuclein, amyloid beta-peptide and mutant huntingtin. Finally, the authors find that worm lifespan can be increased by treating them with a sulfatase inhibitor (STX64) and a learning and memory deficit is ameliorated by the same inhibitor in APP/PS1 mutant transgenic mice.

The findings are very interesting and the experiments appear to be well done. There are, however, several gaps with regards to mechanisms responsible for the phenotypes of the sul-2 mutant worms and in interpretation of some of the data.

1. The authors show that Sul-2 is only expressed in sensory neurons. This raises two major questions that should be addressed.

First, are all the steroids shown in ED Table 1 desulfated in the sensory neurons? This would be remarkable.

Second, what is the steroid(s) that is/are desulfated by Sul-2 that is/are responsible for the longevity phenotype of Sul-2 mutants?

2. What is the mechanism by which Sul-2 deficiency decreases the accumulation alpha-synuclein, A-beta and huntingtin aggregates? If via steroids, which steroid(s).

3. Do Sul-2 mutants have reduced food intake? Perhaps the worms lacking functional Sul-2 in their sensory neurons consume less food/fewer calories and this accounts for their increased longevity (and protection in neurodegenerative disorder models).

4. The manuscript requires considerable editing to correct grammatical errors.

Reviewer #3 (Remarks to the Author):

In this manuscript, Munoz and colleagues investigate the impact of mutated form of the steroid hormone sulfatase SUL-2 on lifespan in the nematode *C. elegans*. They first note that loss-of-function alleles of *sul-2* increase lifespan and show data that support the idea that this is linked to its enzymatic activity as the steroid hormone sulfatase inhibitor STX64 has the same effect. Through epistasis, they show data suggesting that *sul-2* acts in the germline pathways although it is localized in sensory neurons. This observation leads the authors to test the impact of *sul-2* alleles and STX64 on proteostasis in *C. elegans* and finally in neurodegeneration mice models.

My overall impression is that, although there are interesting data in the paper, it is confusing. Lots of questions remain to be addressed. What is the link between the germline and sensory neurons? Is *sul-2* expressed in the somatic gonad? Complementing a *sul-2* null with tissue specific *sul-2* loss-of-function alleles would have been very informative. How does the sulfatase activity mediate the lifespan and proteostasis effect? What about the role of other sulfatases?

I am not saying that all of these points should be tackled but choosing one angle and digging deeper would have been nice and the paper may, at the end, seem more consistent.

We are left with bits and pieces of a great puzzle that leaves a feeling of unfinished work.

One technical comment: were the lifespan experiments performed at 25 degrees? If so, why? Most published lifespan data is done at 20. For the sake of comparison, it would have been better to stick to standard temperature.

Figure 1 : *data using STX64 is weak. As it is normally done with drugs, the authors should show a dose dependent response.

*it is regrettable that the authors only show percentage increase when it comes to sulfated steroid hormones. They should show absolute value. This will give the reader a better understanding of the role of SUL-2

Figure 2 : *to do these epistasis test, I would strongly recommend using the same allele all the way to ease the interpretation of the data. Only in case where the use of an allele is made impossible for reasons that should be explained can this happen. The reader should be alerted about the rationale of using an allele or the other is there is one.

*the epistasis analysis is pretty consistent with the exception of nhr-80, which is interesting and fine to me. But does not this beg testing nhr-49 that is also implicated in the pathway and seems to work “with” nhr-80?

* a genetically clean experiment to test the interaction of sul-2 and DA would have been to compare the effect of DA supplementation on the lifespan of glp-1;daf-12 and glp;daf-12;sul-2. The same can be done with daf-36.

Figure 3 : *although I do not work specifically with neurodegeneration or proteostasis, my impression is that most (if not all?) alleles that extend lifespan positively affect proteostasis. I would like to know what the authors think about this. Do these data merely reflect the effect of sul-2 on lifespan or is there any argument that its effect is somewhat age-dependent?

Figure 4: I do not feel competent to comment this figure as this is not my field of expertise

Tipo :

Line 61: Hormone dependending cancers should read Hormone dependent cancers

Suggestion:

Line 172: I would suggest: Data from many labs suggest that steroid hormones are implicated in this process. Our data confirm this idea.

Hugo Aguilaniu

We would like to take the opportunity to thank the reviewers for a careful and constructive criticism of our manuscript. We are pleased with their largely positive response and have endeavoured to further improve the manuscript in light of their observations.

Below, we address each of the points of the reviewers and state what changes have been made in response to them. Changes are indicated by underlining.

Sincerely yours,

Manuel J. Muñoz

Reviewers' comments:

Reviewer #1 (Remarks to the Author):

In the manuscript, Perez-Jimenez and colleagues launched an investigation of the role of *C. elegans* sul-2 in longevity. The authors reported that loss of function of sul-2 led to increased lifespan in worms. Since sul-2 encodes a sulfatase, the authors hypothesized that sul-2 promoted longevity by modifying sulfated steroid hormones. In support of their hypothesis, the authors showed that pharmacological inhibition of sulfatase activity with a human cancer drug STX64 increased sulfated hormones and reproduced the longevity phenotype in worms. These authors continued to use sophisticated genetic tools to demonstrate that sul-2-mediated increase in longevity resembled that in mutant worms lacking germline signaling. Surprisingly, sul-2 deficiency didn't affect fertility, reproductive age or gonadal function, suggesting that one or more of the sul-2 targets can modulate longevity without impairing reproductive functions. Finally, the authors found that sul-2 deficiency improved symptoms of neurodegenerative models in worms and in mice. The research subject is of great interest. Moreover, the findings that targeting sul-2 pathway can be beneficial in worm and murine models of neurodegenerative diseases are quite novel and potentially important. That said, the current manuscript has its limitations.

First, without the knowledge of the specific steroid hormones underlying the increased longevity, the study lacks mechanistic insight. It is particularly hard to interpret how sul-2 expression in few sensory neurons could impact longevity at the organismal level.

The comments of the reviewers encouraged us to search for the causative hormone. We have found that the steroid hormone responsible for the improvement of the protein aggregation models symptoms are sulfated C19 androgen steroids hormones. We observed that sulfated testosterone, sulfated epitestosterone and sulfated dehydroepiandrosterone improve symptoms in one or several models of protein aggregation disease and specially sulfated epitestosterone which is able to improve all the models tested and also increases longevity in a wild type background. No effect is observed with the non-sulfated form or with sulfated C21 steroid hormone either. In addition, inhibition of the synthesis of C19 steroids hormones suppresses the benefit of the mutation in the sulfatase gene. With these results, a paragraph is included in the manuscript describing the experiments that demonstrate the beneficial effect of the sulfated C19 androgen steroid

hormones, a full figure (figure 4) is incorporated in the manuscript and the model has changed to incorporate this new finding.

Does sul-2 homolog in mammals have a similar expression pattern in the nervous system?

STS human homologue of sul-2, is described to be expressed in different tissues noteworthy it is strongly expressed in the nervous system. We acknowledge this suggestion and agree that this observation enhances our findings giving a better view of a conserved process. We have included a sentence in the discussion where this is highlighted and a bibliographic reference is also included.

The authors cited one paper of STS deficiency in humans. Is there any evidence that such a deficiency in humans or in rodents increases a healthy lifespan?

This is also a very interesting suggestion. We did not find any data in literature that indicate an increase of healthy lifespan or reduced risk of neurodegenerative diseases in those patients. We are planning to collaborate with Medical Doctors to find out if any aging markers could be altered in humans with STS deficiency.

Second, lifespan is used as a functional readout for most of the genetic experiments without necessary information regarding altered physiologies (e.g. metabolism, intake, mobility) leading towards longevity.

We have done several studies to evaluate the physiology of the aging process in the sul-2 mutant background. We have analyzed mobility and pharyngeal pumping frequency. This new data are included in supplementary figure 1.

Finally, sul-2's effects on longevity and neurodegenerative diseases could involve independent steroid hormones and pathways. Therefore, the findings that inhibiting sul-2 ameliorated symptoms of neurodegenerative models could be the subject of an independent study. Although the data presented in the paper are quite exciting, the therapeutic potential of the drugs like STX64 is dampened by the concern that multiple steroid hormones are affected producing both beneficial and untoward effects.

We agree, treatment with STX64 could alter an unknown group of hormones and some of them might have a negative effect in patients. This situation should be studied carefully before clinical application. Interestingly, STX64 has been already used in phase I and phase II clinical assays for hormone dependent cancer. Neither phase I nor phase II studies reported serious secondary effects, which encourage the possibility of using it as therapeutic for at least serious diseases as neurodegeneration. We have including a sentence in the discussion indicating this fact.

Reviewer #2 (Remarks to the Author):

By quantifying lifespan in *C. elegans* in which the steroid sulfatase sul-2 gene is

disabled, the authors provide evidence that sul-2 limits lifespan. Interestingly, sul-2 is expressed only a few specific sensory neurons suggesting the desulfation of one or more steroids in those neurons is responsible for limiting lifespan. The authors also find that sul-2 mutant worms are protected against proteotoxicities of alpha-synuclein, amyloid beta-peptide and mutant huntingtin. Finally, the authors find that worm lifespan can be increased by treating them with a sulfatase inhibitor (STX64) and a learning and memory deficit is ameliorated by the same inhibitor in APP/PS1 mutant transgenic mice.

The findings are very interesting and the experiments appear to be well done. There are, however, several gaps with regards to mechanisms responsible for the phenotypes of the sul-2 mutant worms and in interpretation of some of the data.

1. The authors show that Sul-2 is only expressed in sensory neurons. This raises two major questions that should be addressed.

First, are all the steroids shown in ED Table 1 desulfated in the sensory neurons? This would be remarkable.

This observation is very interesting. We could not answer this specific question. Steroid hormones are synthesized in a non-sulfated form and then, they can be sulfated and then desulfated. From the pool of non-sulfated hormones, we cannot distinguish those that are desulfated by sulfatases or just newly synthesized. Probably only a small percentage would be desulfated.

Second, what is the steroid(s) that is/are desulfated by Sul-2 that is/are responsible for the longevity phenotype of Sul-2 mutants?

The identification of the hormone responsible of the phenotypes has been our main goal. We have found that treatment with epitestosterone sulfate mimics the increased longevity observed in sul-2 mutant. Other phenotypes described for the mutant are also observed upon treatment with this sulfated steroid hormone. The data we showed suggest that a sulfated hormone, that probably is epitestosterone sulfate or a close related hormone, should be responsible of the increased longevity of the sul-2 mutant. A full paragraph about the finding of this hormone has been incorporated in manuscript and discussion. The model has been changed and also the title of the manuscript. We acknowledge the reviewer's comments that encourage us to work in this direction, helping to clarify how sul-2 regulates longevity and protein aggregation.

2. What is the mechanism by which Sul-2 deficiency decreases the accumulation alpha-synuclein, Abeta and huntingtin aggregates? If via steroids, which steroid(s).

We have found that the steroid hormone responsible of the improvement of the symptoms of the protein aggregation models are sulfated C19 androgen steroid hormones. We observed that sulfated testosterone, sulfated epitestosterone and sulfated dehydroepiandrosterone improve one or several models of protein aggregation disease and specially sulfated epitestosterone, which is able to improve all the models tested and also increases longevity in a wild type background.

No effect is observed in the non-sulfated form neither in sulfated C21 steroid hormone. In addition, inhibition of the synthesis of C19 steroids hormones suppresses the benefit of the mutation in the sulfatase gene.

The presence of this sulfated C19 androgen steroids hormone should activate the same factors that germ-line deficient animals, a condition which in fact have also demonstrated a neuroprotection effect. A paragraph has been included in the discussion.

3. Do Sul-2 mutants have reduced food intake? Perhaps the worms lacking functional Sul-2 in their sensory neurons consume less food/fewer calories and this accounts for their increased longevity (and protection in neurodegenerative disorder models).

In light of this comment, we have studied pharyngeal pumping in the mutant, as a measure of food intake. The longevity or protection in neurodegenerative disorders does not correlate with reduction of food intake. We also observe that dietary restriction further increases the life span of sul-2 mutants, suggesting that reduction of food intake is not the cause of its increased longevity. We have incorporated this data in figure 2k and extended data figure 1d.

4. The manuscript requires considerable editing to correct grammatical errors.

Two scientists whose have reviewed the manuscript with special attention to the English grammar.

Reviewer #3 (Remarks to the Author):

In this manuscript, Munoz and colleagues investigate the impact of mutated form of the steroid hormone sulfatase SUL-2 on lifespan in the nematode *C. elegans*. They first note that loss-of-function alleles of *sul-2* increase lifespan and show data that support the idea that this is linked to its enzymatic activity as the steroid hormone sulfatase inhibitor STX64 has the same effect. Through epistasis, they show data suggesting that *sul-2* acts in the germline pathways although it is localized in sensory neurons. This observation leads the authors to test the impact of *sul-2* alleles and STX64 on proteostasis in *C. elegans* and finally in neurodegeneration mice models.

My overall impression is that, although there are interesting data in the paper, it is confusing. Lots of questions remain to be addressed.

What is the link between the germline and sensory neurons? Is *sul-2* expressed in the somatic gonad? Complementing a *sul-2* null with tissue specific *sul-2* loss-of-function alleles would have been very informative.

I am not saying that all of these points should be tackled but choosing one angle and digging deeper would have been nice and the paper may, at the end, seem mor consistent.

We are left with bits and pieces of a great puzzle that leaves a feeling of unfinished work.

We agree with the reviewer that the status of the previous manuscript leaved the feeling of unfinished work, with many different pieces unmatched and we needed to choose a precise angle. Main effort was made to focus in understanding the role of the hormones in the sul-2 mutants. Previously, we did not know the function of the sulfated hormones in longevity and neurodegenerative disorder protection observed in the steroid sulfatase sul-2 mutant background. This could be due to reduction of specific hormone/s by sulfate inactivation or by the presence of sulfated hormone which may have an active function (for instance, like modulator of neurotransmission as described).

After treating with different hormones, we found that the steroid hormones responsible of the improvement of the protein aggregation models symptoms are sulfated C19 androgen steroids hormones. We observed that sulfated testosterone, sulfated epitestosterone and sulfated dehydroepiandrosterone improve one or several models of protein aggregation disease and specially sulfated epitestosterone which is able to improve all the models tested and also increases longevity in a wild type background. No effect is observed in the non-sulfated form neither in sulfated C21 steroid hormone. In addition, inhibition of the synthesis of C19 steroid hormones suppresses the benefit of the mutation in the sulfatase gene.

All this data indicate that the presence of one or several type of sulfated C19 androgen steroid hormones is the cause of the beneficial effect observed in the sul-2 mutant. We measure the level of sulfated steroid hormone in glp-1 mutant and we did not observe an increase in the level of sulfated steroid hormone. This generate a profound change in our working model. Previously we favored the idea that gonads probably produce steroid hormones negatively affecting longevity. Germline less animals should reduce the pool of hormones produced and this could be the reason for their increased longevity. In sul-2 mutants the same outcome could be obtained through inactivation of this type of hormone by keeping them in a sulfated state. The fact that sulfated steroid hormones generate the beneficial effect changed completely our point of view and indicated that this hypothesis was wrong.

Our idea now is that gonads probably produce steroid hormones that can be change to their sulfated forms by sulfotransferases. These sulfated hormones induce a protective signal that shares most of the elements for germline-less longevity pathway. The fact that sulfotransferase and sulfatase are mainly expressed in sensory neurons suggests that they could be sensing environmental factors coordinating gonadal state with environmental signals.

We do not observe expression of sul-2 in any other tissue neither when expressed from the chromosomal array or from the integrated strain. It would be very nice to test if the sensory neurons are involved in the sulfation of the hormones by complementation with sul-2 as suggested, but unfortunately we could not afford this experiment because we were focused in this other issue. We expect to address that in future works together with the role of the sulfotransferase in regulation of longevity.

One technical comment: were the lifespan experiments performed at 25 degrees? If so, why? Most published lifespan data is done at 20. For the safe of comparison, it would have been better to stick to standard temperature.

We started to work in a fer-15(b26) background. This mutation produces no active sperm at 25°C, and animals are therefore sterile. This situation facilitates the longevity

assay. Taking in consideration this concern we have repeated several experiments at 20°C, to see consistency of the longevity result, observing similar results at both temperatures. Those experiments are shown on sul-2 mutants, in extended figure 1b and some of the experiment with STX64. In the other experiment with double mutants we checked that every simple mutant behaves as described.

Figure 1 : *data using STX64 is weak. As it is normally done with drugs, the authors should show a dose dependent response.

We have included a dose dependent response in extended data Fig. 3. Repetition is included in supplementary table 1

*it is regrettable that the authors only show percentage increase when it comes to sulfated steroid hormones. They should show absolute value. This will give the reader a better understanding of the role of SUL-2

We agree that absolute values are better, but it is complex from the technical point of view, as it includes calibration of the different hormones. We believed that for comparison purposes, to test different level between wild type and mutants was enough.

Figure 2 : *to do these epistasis test, I would strongly recommend using the same allele all the way to ease the interpretation of the data. Only in case where the use of an allele is made impossible for reasons that should be explained can this happen. The reader should be alerted about the rationale of using an allele or the other is there is one.

We have changed to show the result with the deletion allele in the main figure 2 and the results with the punctual mutation allele in the extended data Fig 4. Deletion allele has the advantage that conclusions are more robust. However, in some experiments it shows a bimodal curve with a subpopulation suffering an early death. Some of the punctual allele data were done after this comment and they show a difference with the deletion allele, suggesting some other effect by the punctual allele. All the new data are included in the manuscript.

*the epistasis analysis is pretty consistent with the exception of nhr-80, which is interesting and fine to me. But does not this beg testing nhr-49 that is also implicated in the pathway and seems to work “with” nhr-80?

We have studied the epistasis analysis with nhr-80, after this suggestion. Interestingly, neither nhr-80 nor nhr-49 suppress the increase of longevity of sul-2, which suggests that sul-2 share many elements of the germline signaling pathway but not all.

* a genetically clean experiment to test the interaction of sul-2 and DA would have been to compare the effect of DA supplementation on the lifespan of glp-1;daf-12 and glp;daf-12;sul-2. The same can be done with daf-36.

The effect of DA in sul-2 is really a very interesting experiment. We know that daf-36 is essential for the increased longevity observed upon sul-2 inhibition and it would be very interesting to test if adding dafachronic acid would restore the increased of longevity in a daf-36 sul-2(-) background. However, we got some technical issues that we could not

overcome during the review process. In our hands, DA increased longevity in a wild type background, this is not described and we need to understand why we got a different result before we go ahead. Another difficulty was that sul-2 is very close in the chromosome to daf-36, so we could not get a double mutant. We tried to inhibit sul-2 with STX64, but using two drugs (STX64 + Dafachronic acid) in the same experiment were not the best conditions. We are now trying to generate a deletion allele of sul-2 by Crispr in a daf-36 mutant background.

Figure 3 : *although I do not work specifically with neurodegeneration or proteostasis, my impression is that most (if not all?) alleles that extend lifespan positively affect proteostasis. I would like to know what the authors think about this. Do these data merely reflect the effect of sul-2 on lifespan or is there any argument that its effect is somewhat age-dependent?

Yes, most of the alleles that increase longevity improve proteostasis. We believe that in a postmitotic organism like C. elegans, the aging process occurs due to the accumulation of damage inside the cells. Loss of proteostasis is a process described in aging nematode which drives to protein aggregation and deterioration. We believe that aging and proteostasis are highly linked processes in C. elegans. We introduced a paragraph in the discussion to highly this view which explains that the full manuscript is about one process, which is proteostasis, with two outcomes, aging in natural proteins and neurodegenerative diseases for ectopic proteins that are prone to aggregate.

Figure 4: I do not feel competent to comment this figure as this is not my field of expertise

Tipo :

Line 61: Hormone dependending cancers should read Hormone dependent cancers

Changed, Thanks

Suggestion:

Line 172: I would suggest: Data from many labs suggest that steroid hormones are implicated in this process. Our data confirm this idea.

We have removed this sentence and included a paragraph with a more extensive discussion

Hugo Aguilaniu

Thank you very much for your comments

Manuel

REVIEWER COMMENTS

Reviewer #1 (Remarks to the Author):

All my concerns have been addressed.

Reviewer #3 (Remarks to the Author):

The manuscript has clearly improved and I congratulate the authors on the work they have produced. The article is more appealing and new data on hormone supplementation are really interesting.

In my view there is one key missing experiment to finish this paper: the authors should measure the effect of ES on the lifespan of sul-2 mutants. one would expect that no effect should be observed.

I strongly recommend doing this last experiment before publishing this work in Nature Communications.

Minor point:

line 142 should read: "was observed, which are also..." instead of "as observed. Which are also..."

Hugo Aguilaniu

We would like to acknowledge all the suggestions of the Reviewers that largely improve the manuscript.

Below, we address each of the points of the Reviewer and state what changes have been made in response to them. Changes are indicated in the manuscript by underlining.

Reviewer 3:

Indicated that “the authors should measure the effect of ES on the lifespan of *sul-2* mutants. One would expect that no effect should be observed.”

We have measure the effect on lifespan of ES treatment in both *sul-2* mutant backgrounds. The results of the experiments showed that ES treatment did not further increase the longevity of *sul-2* mutants. This result indicates that both, increased longevity of *sul-2* and the effect of ES treatment share the same molecular mechanism, as expected by the Reviewer 3. We have included this result in the text in the line 261 and in the figure 4f and a biological replica is included in Supplementary figure 9c as well as in Supplementary Dataset 1 (longevity table).

We have included the minor change of the Reviewer 3 in the line 142.

Thank you for your comments

Manuel